# Equilibrium Pricing in Oligopolistic Data Markets

**Bhaskar Ray Chaudhury** [* 1]  **Jugal Garg** [1]  **Eklavya Sharma** [* 1]  **Jiaxin Song** [* 1]

## Abstract

We study equilibrium pricing in oligopolistic data markets with budget-constrained buyers (e.g., ML companies purchasing data to improve model accuracy) and strategic data sellers. Sellers compete by setting prices for their datasets, giving rise to a pricing game whose pure Nash equilibria correspond to equilibrium prices. While equilibrium prices are guaranteed for rivalrous goods via competitive equilibrium, we show that the non-rivalry of data fundamentally alters this picture: an exact Nash equilibrium (NE) need not exist, and in fact no 1.36-approximate NE exists under uniform pricing. We therefore investigate relaxed equilibrium notions. Allowing sellers to use beyond-uniform pricing—specifically, piecewise-linear convex pricing functions—guarantees approximate stability within a constant factor: there exists a pricing profile in which no seller can improve revenue by a factor of two by deviating to any uniform price (a 2-approximate NE). Finally, our simulations demonstrate fast convergence and empirical approximation guarantees that outperform the worst-case bound of 2.

## 1. Introduction

The rapid growth of data-driven decision-making and the widespread adoption of data-centric technologies have cemented data as one of the most valuable assets of the $21^{st}$ century. Falling storage costs, together with major advances in data mining, analytics, and machine learning, have dramatically increased both the usefulness and the economic value of data. According to Acumen Research (2021), the U.S. big data market is expected to reach approximately $473 billion by 2030, highlighting the growing role of data in shaping innovation, productivity, and market dynamics.

A fundamental question in today's data economy is *"how is data priced by data sellers"?*. In this paper, we study data pricing in *oligopolistic markets*, i.e., markets where a small number of sellers strategically choose prices for their datasets. These seller interactions give rise to a pricing game, and equilibrium prices correspond to a pure Nash equilibrium of this game. This framework provides a natural parallel to data pricing in perfectly competitive markets—where sellers are price takers and prices are determined by equating supply and demand for each dataset—studied in recent work (Chaudhury et al., 2026a).

**Value of data.**    To formalize our framework, we begin by specifying how data generates value for a buyer. In our model, buyers represent AI/ML agents who acquire data with the goal of improving the quality of their predictions. Formally, each buyer $i$, has a maximum budget of $b_i$, and seeks to infer an unknown parameter $\theta_i$, corresponding to an underlying quantity of interest such as future demand, user behavior, or system performance. Individual data records are modeled as digitized observations that provide noisy information about $\theta_i$. Data supplied by a given seller is drawn from a fixed distribution that reflects the seller's domain of activity and the informativeness of the data with respect to $\theta_i$. These distributions may differ across sellers, capturing heterogeneity in both data generation processes and predictive relevance. As an illustration, consider a buyer attempting to predict future demand for a product. One seller may provide historical transaction logs capturing purchase frequency and timing, while another may offer mobility or web traffic data that indirectly correlates with consumer interest. Although both datasets convey information about the same latent parameter, they differ in structure, noise characteristics, and informational content.

Suppose there are $m$ sellers. A buyer's acquisition decision is represented by a *data bundle* $\mathbf{x}_i = (x_{i,1}, x_{i,2}, \ldots, x_{i,m})$, where $x_{i,j}$ denotes the number of data records purchased by buyer $i$ from seller $j$. Seller $j$'s dataset can be viewed as a set of $s_j$ data-records. The full collection of acquired signals, denoted by $S(\mathbf{x}_i)$, consists of one signal per data record. Upon observing these signals, the buyer updates

---

[*]Equal contribution; Bhaskar Ray Chaudhury, Eklavya Sharma, and Jiaxin Song are equal first authors. [1]Department of Industrial and Enterprise Systems Engineering, University of Illinois at Urbana-Champaign, USA. Correspondence to: Bhaskar Ray Chaudhury <braycha@illinois.edu>, Jugal Garg <jugal@illinois.edu>, Eklavya Sharma <eklavya2@illinois.edu>, Jiaxin Song <jiaxins8@illinois.edu>.

*Proceedings of the 43$^{rd}$ International Conference on Machine Learning*, Seoul, South Korea. PMLR 306, 2026. Copyright 2026 by the author(s).

her belief about the latent parameter, yielding a posterior distribution $\theta_i \mid S(\mathbf{x}_i)$. Following the recent literature on the economic value of data (Baley & Veldkamp, 2025), and the recent work on data pricing (Chaudhury et al., 2026a), we define the buyer's *utility* from $\mathbf{x}_i$, denoted by $u_i(\mathbf{x}_i)$, as the resulting reduction in uncertainty about $\theta_i$. Specifically, utility is measured as the expected increase in precision:

$$u_i(\mathbf{x}_i) = \alpha_i \cdot (\mathbb{E}[\mathrm{Pre}(\theta_i \mid S(\mathbf{x}_i))] - \mathrm{Pre}(\theta_i)),$$

where the precision of a random variable is defined as the inverse of its variance, i.e., $\mathrm{Pre}(\theta) = 1/\mathrm{Var}(\theta)$, and $\alpha_i$ is the value per unit increase in precision for buyer $i$.

**Pricing game and Nash equilibria.** Each seller $j$ chooses a price $p_j$ per data record of their dataset. This pricing model reflects common practices in real data marketplaces. For example, commercial platforms such as Snowflake Marketplace allow data providers to charge buyers on a usage-based basis, including per-query and per-row pricing for access to paid dataset (Suger.io Documentation, 2026). Similarly, third-party data marketplaces employ volume-based pricing schemes, with providers such as Bright Data charging per thousand records accessed (Nulty, 2026).

Given a profile of seller prices $\boldsymbol{p} = (p_1, p_2, \ldots, p_m)$, each buyer $i$ demands an affordable bundle $\boldsymbol{x}_i$ that maximizes her utility, i.e, $\boldsymbol{x}_i \in \arg\max_{\boldsymbol{z} \in \mathbb{R}_{\geq 0}^m \mid \boldsymbol{p}^T \boldsymbol{z} \,\leq\, b_i} u_i(\boldsymbol{z})$. When there are multiple optimal bundles, we use a natural tie-breaking rule (see Section 4.2 for details). Therefore, given a price profile $\boldsymbol{p}$, let $\boldsymbol{x}_i^*(\boldsymbol{p})$ denote the optimal demand bundle for buyer $i$. Then, the revenue of seller $j$, $r_j(p_j, \boldsymbol{p}_{-j}) = p_j \cdot \sum_i \boldsymbol{x}_i^*(\boldsymbol{p})$. A price profile $\boldsymbol{p}$ is an equilibrium if no seller can increase her revenue by unilaterally deviating; that is, for every seller $j$ and every alternative price $p_j'$, we have $r_j(p_j, \boldsymbol{p}_{-j}) \geq r_j(p_j', \boldsymbol{p}_{-j})$.

**Economic effects of non-rivalry.** It is well known that in classical markets with *rivalrous goods* and *linear buyer utilities*[1], any *competitive equilibrium* (CE)—that is, a price profile at which each good's aggregate demand equals its supply—is also an oligopolistic equilibrium. In particular, at a CE no seller has an incentive to unilaterally deviate from her price; we provide a brief argument in the full version (Chaudhury et al., 2026b). This equivalence, however, fails to extend to markets with *non-rivalrous* goods, such as data. Indeed, one can show that the notion of competitive equilibrium introduced in (Chaudhury et al., 2026a) does not, in general, constitute a Nash equilibrium (NE) in oligopolistic data markets—standing in sharp contrast to the classical rivalrous setting. These observations underscore that classical CE theory cannot be directly applied to oligopolistic markets with non-rivalrous goods, necessitating a separate

___
[1] A good is rivalrous if its availability to one buyer is affected by its consumption by other buyers.

analysis of equilibrium in data markets, which is the focus of this paper.

## 1.1. Our Contributions

Consistent with (Chaudhury et al., 2026a), for all our results, we assume that for each buyer $i$, $\theta_i \sim \mathcal{N}(0, \tau_i^{-1})$. Each data-record of seller $j$ is a signal $s_{i,j} = \theta_i + \eta_{i,j}$ to buyer $i$, where $\eta_{i,j} \sim \mathcal{N}(0, \tau_{i,j}^{-1})$. It can be shown that under the foregoing assumptions, we have $u_i(\boldsymbol{x}_i) = \sum_j \tau_{i,j} x_{i,j}$. We now list our main contributions.

1. We show that there exist instances of oligopolistic data markets that admit no equilibrium (Section 3)—this is a sharp contrast to rivalrous oligopolistic markets. We also manage to find instances that admit no 1.36 Nash equilibrium (Theorem 3.2).

2. We investigate natural relaxations of NE by allowing sellers to adopt more flexible pricing strategies. In particular, instead of assigning a uniform price to every data record, we allow sellers to use *piecewise-linear convex* (PLC) pricing functions, which have been shown to be revenue-optimal for a monopolist selling data to heterogeneous buyers (Chaudhury et al., 2025).

   Unfortunately, even within the broader class of PLC pricing strategies, a pure NE may fail to exist (Theorem 3.2). Further, computing an optimal PLC strategy is computationally hard (see the full version (Chaudhury et al., 2026b) for the proof), suggesting that sellers may be expected to respond with coarse PLC pricing strategies or even uniform pricing, rather than optimizing over the full PLC space.

   Motivated by this observation, we relax the equilibrium concept by restricting the class of deviations, rather than the strategy space itself. Specifically, we prove the existence of a PLC price profile such that no seller can deviate to a uniform pricing strategy and earn more than twice her current revenue, providing a meaningful approximate equilibrium despite the nonexistence of an exact one (Theorem 4.1). This viewpoint parallels classical ideas in game theory and learning, where outcomes generated by rich or adaptive strategies are evaluated relative to a simpler benchmark class—most notably in regret minimization, where performance is compared against the best fixed strategy in hindsight. Here, uniform pricing serves as a natural behavioral baseline, grounding the relaxation in both theory and practice.

3. Finally, we run some empirics to show the robustness of our results. We simulate a *news data marketplace* where $n$ firms purchase data from $m$ news agencies to predict stock price movements, from a real-world dataset (Aaron7sun, 2016). Each firm evaluates the informativeness of a dataset by training a logistic regression

model on historical news and measuring prediction error. Sellers update their pricing strategies via an *inertial approximate best-response dynamics* (defined in Section 5).

Empirically, the dynamics converge efficiently across all settings, exhibiting a non-monotone pattern: convergence is slower when the number of sellers is very small due to strong competition effects, improves sharply as the market grows, and then increases steadily in larger markets, reflecting a large-market stabilization effect. Secondly, although theoretical guarantees apply only to approximate equilibria, the dynamics reach an exact Nash equilibrium in all our simulations, highlighting a gap between worst-case bounds and typical behavior and motivating further theoretical investigation into the structural properties of real-world pricing games. Overall, our empirical findings reveal several intriguing behaviors that merit more rigorous investigation in future work.

**Beyond Linear Utilities.** Our results also work for slightly more general buyers' utility functions: $u_i(\boldsymbol{x}_i) = g_i\left(\sum_j \tau_{i,j} x_{i,j}\right)$, where $g_i : \mathbb{R}_{\geq 0} \to \mathbb{R}_{\geq 0}$ is an increasing continuous function such that $g_i(\boldsymbol{0}) = 0$. This is because any monotonic transformation of a buyer's utility function does not change her set of utility-maximizing bundles. When $g_i$ is concave, this formulation captures diminishing marginal returns of data.

This more general form also captures other ways to model the value of data. Instead of defining the value of data as the *expected increase in precision*, we can also define it as the *expected reduction in variance*, i.e.,

$$u_i(\boldsymbol{x}_i) = \alpha_i \cdot (\mathrm{Var}(\theta_i) - \mathbb{E}[\mathrm{Var}(\theta_i \mid S(\boldsymbol{x}_i))]).$$

Assuming $\theta_i$ and $\eta_{i,j}$ to be normally-distributed as before, we get

$$u_i(\boldsymbol{x}_i) = \alpha_i \cdot \left(\frac{1}{\tau_i} - \frac{1}{\tau_i + \sum_{j=1}^{m} \tau_{i,j} x_{i,j}}\right).$$

One can verify that this expression is concave in $\sum_{j=1}^{m} \tau_{i,j} x_{i,j}$. Alternatively, we can define the value of data as the *expected reduction in entropy*. This gives us

$$u_i(\boldsymbol{x}_i) = \alpha_i \cdot \left(\ln\left(\tau_i + \sum_{j=1}^{m} \tau_{i,j} x_{i,j}\right) - \ln(\tau_i)\right).$$

This expression is also concave in $\sum_{j=1}^{m} \tau_{i,j} x_{i,j}$.

Since the choice of $g_i$ doesn't affect buyer behavior, we can let $g_i$ be the identity function without loss of generality. This makes $u_i$ linear, which simplifies exposition.

## 1.2. Related Work

The economics of data has attracted growing attention due to the widespread adoption of AI and data-centric technologies. While a full overview is beyond the scope of this paper,

we focus on the literature most closely related to our study. We adopt a general model of data value, where a buyer's utility is measured by the improvement in prediction accuracy. Prior work has explored more specialized valuation frameworks (Farboodi et al., 2025; Veldkamp, 2023; Farboodi & Veldkamp, 2023). We refer the reader to (Fleckenstein et al., 2023) for a detailed review of data valuation methods.

A significant line of research studies mechanisms that incentivize agents to share data, often by compensating for privacy loss (Fallah et al., 2024; Cummings et al., 2023; Fallah et al., 2022; Murhekar et al., 2023). There have been studies on mechanisms that incentivize sellers to truthfully report the variances of their datasets to a data aggregator, who aims to achieve a target prediction accuracy (Cummings et al., 2015).

Data markets, which match buyers' prediction requests with datasets from sellers, have been studied extensively in terms of strategic behavior, incentives, and revenue optimization. Monopolist revenue-maximization strategies have been analyzed (Admati & Pfleiderer, 1986; 1990; Bergemann et al., 2018; Babaioff et al., 2012). (Agarwal et al., 2019) designs a truthful mechanism where buyers pay in proportion to the accuracy gains they receive, yielding a discriminatory but explicit pricing scheme. Other work investigates equilibria and auctions in data markets with externalities (Agarwal et al., 2024; Hossain & Chen, 2024), first-principles approaches to data pricing (Mehta et al., 2021; Pei, 2020; Cai & Velegkas, 2021; Bergemann et al., 2022), and stable outcomes in non-monetary data exchange economies (Bhaskara et al., 2024; Akrami et al., 2025; Song et al., 2025).

## 2. Preliminaries

Let $[t]$ denote the set $\{1, 2, \ldots, t\}$ for any $t \in \{0\} \cup \mathbb{N}$. For any $m \in \mathbb{N}$, define the simplex $\Delta_m := \{\boldsymbol{x} \in \mathbb{R}_{\geq 0}^m : \sum_{j=1}^m x_j = 1\}$. For any $\boldsymbol{x} \in \mathbb{R}^m$, let $\mathrm{supp}(\boldsymbol{x}) := \{j \in [m] : x_j \neq 0\}$. For any $m \in \mathbb{N}$, let $\boldsymbol{0}^{(m)}$ and $\boldsymbol{1}^{(m)}$ denote the $m$-dimensional vectors of all zeros and all ones, respectively. When $m$ is clear from context, we simply write $\boldsymbol{0}$ and $\boldsymbol{1}$. For any $j, m \in \mathbb{N}$, let $\boldsymbol{e}^{(j,m)}$ denote an $m$-dimensional vector whose $j^{\text{th}}$ component is 1 and all other components are 0. When $m$ is clear from context, we write $\boldsymbol{e}^{(j)}$.

A data marketplace instance is given by the tuple $([n], [m], (u_i)_{i=1}^n, (b_i)_{i=1}^n)$. Here $[n]$ is the set of buyers and $[m]$ is the set of sellers. Each buyer $i$ has a budget $b_i \in \mathbb{R}_{\geq 0}$ and a utility function $u_i : [0, 1]^m \to \mathbb{R}_{\geq 0}$. Given a bundle $\boldsymbol{z} = (z_1, \ldots, z_m)$, where $z_j$ is the fraction of dataset $j$, buyer $i$ has a value of $u_i(\boldsymbol{z})$ for it.

Based on our model of the value of data (see Section 1), the utility function $u_i$ is linear *linear*. Specifically, each buyer $i$ has value $\tau_{i,j}$ for each dataset $j$, and $u_i(\boldsymbol{z}) = \sum_{j=1}^m \tau_{i,j} z_j$.

There are many ways in which sellers can price their datasets. A dataset with price $p$ is said to be priced uniformly/linearly if an $x$-fraction of the dataset costs $px$.

## 2.1. Buyer Behavior for Linear Pricing

Suppose each seller $j \in [m]$ sets a linear price of $p_j$ for her dataset. Each buyer $i \in [n]$ would like to purchase a bundle of datasets that maximizes her utility subject to her budget constraint. If buyer $i$ purchases an $x_{i,j}$ fraction of each dataset $j$, then she would like to maximize $\sum_{j=1}^{m} \tau_{i,j} x_{i,j}$ under the constraint $\sum_{j=1}^{m} p_j x_{i,j} \leq b_i$. This is the fractional knapsack problem, whose solution is well-understood. Define buyer $i$'s *bang-per-buck* for dataset $j$ to be $\tau_{i,j}/p_j$. To maximize utility, the buyer will first sort the datasets in non-increasing order of bang-per-buck. Let $(\sigma_{i,1}, \ldots, \sigma_{i,m})$ be this ordering. She would then buy as much of dataset $\sigma_{i,1}$ as possible, then buy as much of dataset $\sigma_{i,2}$ as possible, and so on, till she either exhausts her budget or purchases everything.

A slight exception to the above behavior is when multiple datasets have the same bang-per-buck for a buyer. For example, if $\sigma_{i,1}$ and $\sigma_{i,2}$ have the same bang-per-buck for buyer $i$, and her budget is less than $p_{\sigma_{i,1}} + p_{\sigma_{i,2}}$, then any way of distributing her budget across these two datasets is utility-maximizing behavior. We assume that each buyer's *tie-breaking rule*, i.e., how she distributes her budget across multiple datasets having the same bang-per-buck, is known to all sellers. This is necessary to ensure that the pricing game the sellers engage in is a perfect-information game.

## 3. Non-existence of (Approximate) NE

In this section, we show that there exist data market instances for which no (approximate) Nash equilibrium (NE) exists when sellers use linear pricing.

We begin by describing a family of data market instances.

*Example* 3.1. Let $\mathcal{I}$ be a data market instance with two sellers and $n$ buyers. There are two types of buyers: $n-1$ poor buyers and one rich buyer. Let $\alpha$ and $\beta$ be constants such that $1 < \alpha \leq \beta$. Each poor buyer has a budget of 1, and her value for the two datasets are $\tau_{1,1} = \alpha$ and $\tau_{1,2} = 1$, respectively. The rich buyer has a budget of $\beta$ and her values for the two datasets are $\tau_{2,1} = 1$ and $\tau_{2,2} = 0$, respectively.

Since the rich buyer values dataset 2 at zero, she will spend her budget only on dataset 1. The poor buyers consider dataset 1 to be $\alpha$ times as valuable as dataset 2. Thus, if dataset 1 costs more than $\alpha$ times dataset 2, they will prioritize purchasing 2. Similarly, if it costs less than $\alpha$ times dataset 2, they will prioritize 1. (For now, we assume that if dataset 1 costs exactly $\alpha$ times dataset 2, buyers prefer 1. This assumption is relaxed in the full version (Chaudhury et al., 2026b).)

The sellers engage in a pricing game with each other. A strategy profile for this game is given by $(p, q)$, where $p$ and $q$ are the prices of datasets 1 and 2, respectively. We say that $(p, q)$ is a $c$-approximate NE for the pricing game if no seller can increase her revenue by more than a factor of $c$ by changing her dataset's price. We show that, for a suitable choice of $\alpha$ and $\beta$, a 1.363-approximate NE does not exist.

## 3.1. Preliminary Observations

First, let us build some intuition on why a $c$-approximate NE may not exist when $c$ is very close to 1 and $\alpha = \beta = n-1 \geq 3$. Note that seller 1 has no incentive to price above $\beta$, since no buyer can pay more than $\beta$, and seller 2 has no incentive to price above 1, since the rich buyer is not interested, and no poor buyer can pay more than 1. For ease of exposition, we therefore assume $p \leq \beta$ and $q \leq 1$.

**Undercuts must be close.** If $q < p/\alpha$, then seller 2's revenue is $(n-1)q$, which is increasing in $q$. When $p \leq \alpha q$, then seller 1's revenue is $p + (n-1)\min(p,1)$, which is increasing in $p$. Thus, whoever is undercutting the other will do so using the maximum possible price. Thus, if $(p, q)$ is a $c$-approximate NE, then $p \approx \alpha q$.

**For large $p$, undercutting helps significantly.** If $p \geq 1$ and $p \leq \alpha q$, then decreasing dataset 2's price to slightly less than $p/\alpha$ increases seller 2's revenue from 0 to $\approx (n-1)(p/\alpha)$. If $p > \alpha q$, then decreasing dataset 1's price to slightly less than $\alpha q$ increases seller 1's revenue from $\approx \alpha q + (n-1)(1-q)$ to $\approx \alpha q + (n-1)$. Thus, when $p \geq 1$, then some seller can always improve her revenue significantly by undercutting the other, so $(p, q)$ is not a $c$-approximate NE.

**For small $p$, seller 1 raises prices significantly.** If $p \leq 1$ (and so $q \leq 1/\alpha$), then seller 1's revenue is at most $n$. However, if she increases dataset 1's price to $\beta$, her revenue is at least $\beta + (n-1)(1-q) \geq 2n - 3$. Thus, when $p$ is small, seller 1 can always improve her revenue significantly, so $(p, q)$ is not a $c$-approximate NE.

Thus, for $c$ close to 1, $\alpha = \beta = n - 1 \geq 3$, $p \leq \beta$, and $q \leq 1$, $(p, q)$ is not a $c$-approximate NE.

## 3.2. Stronger Inapproximability

We now build on and formalize the ideas from Section 3.1 to obtain a stronger inapproximability result.

**Theorem 3.2.** *In the data market instance of Example 3.1, if we set $\alpha = 0.733(n-1)$, $\beta = 0.860(n-1)$, and $n \to \infty$, then a 1.363-approximate Nash equilibrium does not exist.*

*Proof sketch.* Let $r_1(p, q)$ and $r_2(p, q)$ denote the revenue earned by sellers 1 and 2, respectively, when seller 1 prices her dataset at $p$ and seller 2 prices her dataset at $q$. Let $r_1^*(q)$ be the maximum revenue seller 1 can earn when seller 2 prices her dataset at $q$, and $r_2^*(p)$ be the maximum revenue seller 2 can earn when seller 1 prices her dataset at $p$. To show that a $c$-approximate NE does not exist, we must show that for all $p$ and $q$, either $r_1^*(q) > cr_1(p, q)$ or $r_2^*(p) > cr_2(p, q)$. Equivalently, if we define

$$\mu(p, q) := \max\left(\frac{r_1^*(q)}{r_1(p, q)}, \frac{r_2^*(p)}{r_2(p, q)}\right),$$

then it suffices to prove that $\inf_{p,q} \mu(p, q) > c$.

In the full version (Chaudhury et al., 2026b), we give closed-form expressions for $r_1(p, q)$, $r_2(p, q)$, $r_1^*(q)$, and $r_2^*(p)$, and then minimize $\mu(p, q)$ over all $p, q \in \mathbb{R}_{\geq 0}$. This gives the desired inapproximability factor. $\square$

# 4. PLC Pricing and Approximate NE

In Section 3, we showed that an approximate NE may not exist when sellers use linear pricing. Thus, we investigate a more flexible way to price datasets: *piecewise-linear convex (PLC) pricing*. PLC pricing functions are natural in data market settings, where a seller can divide a large dataset into multiple *shards* and price each shard linearly, as explained below. PLC pricing has also been shown to be revenue-optimal for a monopolist selling data to heterogeneous buyers (Chaudhury et al., 2025).

## 4.1. PLC Pricing and Sharding

We assume that each seller $j \in [m]$ specifies a piecewise-linear and convex *pricing function* $p_j : [0, 1] \to \mathbb{R}_{\geq 0}$, such that $p_j(0) = 0$. Then, the price of purchasing a fraction $x$ of dataset $j$ is $p_j(x)$.

Formally, let there be $m_j$ shards of dataset $j$, where the $k^{\text{th}}$ shard has size $\ell_{j,k}$ and price $p_{j,k}$. The shard sizes sum to 1, i.e., $\sum_{k=1}^{m_j} \ell_{j,k} = 1$. If a buyer purchases $z$ fraction of shard $k$, she would have to pay $p_{j,k}\ell_{j,k}z$.

Without loss of generality, assume $0 < p_{j,1} < p_{j,2} < \ldots < p_{j,m_j}$. Since the shards' prices are increasing, buyers will purchase the shards in order (i.e., they will first purchase shard 1, then shard 2, and so on). Thus, if a buyer purchases a fraction $x$ of the dataset, she would pay $p_{j,1}x$ if $x \leq \ell_{j,1}$, pay $p_{j,1}\ell_{j,1} + p_{j,2}(x - \ell_{j,1})$ if $0 \leq x - \ell_{j,1} \leq \ell_{j,2}$, and so on. Thus, if

$$\sum_{t=1}^{k-1} \ell_{j,t} \leq x \leq \sum_{t=1}^{k} \ell_{j,t}$$

for some $k \in [m_j]$, then

$$p_j(x) = \sum_{t=1}^{k-1} p_{j,t}\ell_{j,t} + p_{j,k}\left(x - \sum_{t=1}^{k-1} \ell_{j,t}\right).$$

Linear pricing is a special case of PLC pricing where there is only a single shard.

Once prices are fixed, each buyer chooses a bundle that maximizes her utility. For linear prices, we showed in Section 2.1 that each buyer faces a fractional knapsack problem, which she solves by ordering the datasets in non-increasing order of bang-per-buck and purchasing greedily. Similarly, under PLC pricing, buyers order the *shards* in non-increasing order of bang-per-buck, and purchase greedily.

## 4.2. 2-NE for PLC Pricing and Linear Deviations

Given the non-existence of (approximate) NE for linear pricing, as shown in Section 3, a natural question is whether NE exist under PLC pricing. In the full version (Chaudhury et al., 2026b), we answer this question negatively: NE may fail to exist even when sellers use PLC pricing.

At a Nash equilibrium under PLC pricing, no seller can increase her revenue by switching to a different PLC pricing strategy. However, we show that computing a seller's best response under PLC pricing is NP-hard; see the full version (Chaudhury et al., 2026b). Consequently, even if a pricing profile is not a NE, sellers may find it difficult to deviate.

Thus, we ask whether there exist pricing profiles that are at least stable against deviations to linear prices. This guarantee is reminiscent of results in game theory, particularly regret minimization, where an adaptive strategy—more general than a fixed strategy—is compared to a fixed strategy in hindsight.

**Discretizing prices.** The strategy space for sellers consists of all PLC functions, which is difficult to work with. For ease of exposition, we simplify the strategy space by *discretizing* prices. Formally, for each seller $j \in [m]$, we are given a finite non-empty set $P_j \subset \mathbb{R}_{>0}$, and we require the price of each shard of dataset $j$ to lie in $P_j$. This is a mild restriction, since any PLC function $p_j$ can be approximated arbitrarily closely by choosing a sufficiently large $P_j$. Moreover, in many real-world settings, prices are typically expressed as integer multiples of a base currency, e.g., $P_j = \{1, 2, \ldots, 10^6\}$, meaning that each shard's price-per-unit must be a whole number of dollars and cannot exceed one million dollars.

The ideas in Section 3.1 carry over to discretized linear prices, provided that the set of prices is sufficiently fine-grained.

Despite these negative results, we show that there exists a

strategy profile for sellers under discretized PLC pricing in which no seller can increase her revenue by more than a factor of two by deviating to a linear pricing strategy.

**Strategy profile.** Let $P_j = \{p_{j,1}, \ldots, p_{j,m_j}\}$ for each seller $j$, where $0 < p_{j,1} < \ldots < p_{j,m_j}$. Seller $j$'s pricing strategy can be described in terms of the shard lengths associated with each price. Formally, her strategy is given by the vector $\boldsymbol{\ell}_j = (\ell_{j,1}, \ldots, \ell_{j,m_j}) \in \Delta_{m_j}$, where $\ell_{j,k}$ is the size of the $k^{\text{th}}$ shard, which has price $p_{j,k}$. The strategy profile of all sellers is denoted by $\boldsymbol{\ell} := (\boldsymbol{\ell}_1, \ldots, \boldsymbol{\ell}_m)$, and we write $\boldsymbol{\ell}_{-j} := (\boldsymbol{\ell}_{j'})_{j' \neq j}$ for the strategies of all sellers other than $j$.

**Buyer behavior and revenue.** Let $r_j(\boldsymbol{\ell})$ be the revenue earned by seller $j$ for the strategy profile $\boldsymbol{\ell}$. For each buyer $i \in [n]$, let $\sigma_i$ be her ordering of shards by bang-per-buck, i.e., the $k^{\text{th}}$ shard of dataset $j$ has bang-per-buck $\tau_{i,j}/p_{j,k}$, and $\sigma_i$ contains all pairs $(j, k)$ in non-increasing order of bang-per-buck. We assume that each buyer $i$ uses *sequential tie-breaking*: when multiple shards have the same bang-per-buck, she preferentially allocates her budget to the shard that appears first in $\sigma_i$ (as opposed to, say, purchasing multiple shards partially). This assumption makes the analysis much simpler. This is a mild assumption because prices are discretized, so a slight perturbation of utilities would eliminate ties in bang-per-buck.

We now state the main result of this section.

**Theorem 4.1** (2-NE for PLC pricing and linear deviations). *There exists a strategy profile $\boldsymbol{\ell}^*$ such that no seller can increase her revenue by more than a factor of 2 by deviating to a linear pricing strategy. Formally, for each seller $j \in [m]$, we have*

$$\max_{k \in [m_j]} r_j(\mathbf{e}^{(k)}, \boldsymbol{\ell}^*_{-j}) \leq 2r_j(\boldsymbol{\ell}^*).$$

We prove Theorem 4.1 in two steps. First, we define each seller $j$'s *randomized revenue* $\widehat{r}_j(\cdot)$ and show that there exists a strategy profile $\boldsymbol{\ell}^*$ such that no seller can increase her randomized revenue by deviating to a linear pricing. Second, we show that for any strategy profile $\boldsymbol{\ell}$ and any seller $j$, the actual revenue satisfies $r_j(\boldsymbol{\ell}) \geq \widehat{r}_j(\boldsymbol{\ell})/2$, i.e., the revenue is always at least half of the randomized revenue.

**Randomized revenue.** Consider the hypothetical scenario where seller $j$ prices the dataset linearly, but the price is decided uniformly randomly, whereas the remaining sellers use (deterministic) PLC pricing. Specifically, seller $j$ sets the price of her dataset to $p_{j,k}$ with probability $\ell_{j,k}$ for all $k \in [m_j]$, whereas every other seller $j'$ has a shard of size $\ell_{j',k}$ having price $p_{j',k}$ for all $k \in [m_{j'}]$. Denote seller $j$'s expected revenue by $\widehat{r}_j(\boldsymbol{\ell})$. Then,

$$\widehat{r}_j(\boldsymbol{\ell}) = \sum_{k=1}^{m_j} \ell_{j,k} r_j(\mathbf{e}^{(k)}, \ell_{-j}).$$

Observe that when $\boldsymbol{\ell}_j = \mathbf{e}^{(k)}$, then $r_j(\boldsymbol{\ell}) = \widehat{r}_j(\boldsymbol{\ell})$.

Under randomized revenues, we can show that a Nash equilibrium always exists using Brouwer's fixed-point theorem.

**Lemma 4.2.** *There is a strategy profile $\boldsymbol{\ell}^*$ such that no seller $j$ can increase her randomized revenue by deviating to a linear pricing. Formally, for all $j \in [m]$, we have*

$$\widehat{r}_j(\boldsymbol{\ell}^*) = \max_{k \in [m_j]} r_j(\mathbf{e}^{(k)}, \boldsymbol{\ell}^*_{-j}).$$

The proof is based on well-known techniques (Nash, 1951), so we defer the full proof to the full version (Chaudhury et al., 2026b).

**From randomized to PLC revenue.** We now explore the relationship between PLC and randomized revenue. We show that the revenue of a PLC strategy is at least half that of the corresponding randomized strategy.

**Lemma 4.3.** *For any seller $j \in [m]$ and any strategy profile $\boldsymbol{\ell}$, we have $r_j(\boldsymbol{\ell}) \geq \widehat{r}(\boldsymbol{\ell})/2$.*

*Proof sketch.* Let $\widehat{\rho}_i$ be the expected revenue from buyer $i$ if seller $j$ prices her dataset linearly at $p_{j,k}$ with probability $\ell_{j,k}$ for each $k$. Let $\rho_i$ be the expected revenue from buyer $i$ if seller $j$ has a shard of price $p_{j,k}$ and length $\ell_{j,k}$ for each $k$. One can show that $\rho_i \geq \widehat{\rho}_i/2$. We defer the full proof to the full version (Chaudhury et al., 2026b). $\square$

By combining Lemmas 4.2 and 4.3, we get the proof of Theorem 4.1.

## 5. Empirics

In this section, we investigate how to find an approximate Nash equilibrium on a *news data market* constructed from a real-world dataset (Aaron7sun, 2016)

**Empirical Setup: Datasets and Buyer Valuations.** The dataset (Aaron7sun, 2016) consists of news headlines paired with indicators of subsequent stock price movements over time. To mimic the interaction between data sellers and buyers, we partition the dataset into two disjoint subsets: one used to simulate the datasets offered by sellers, and another used to simulate the buyer-side test data used for evaluation. In each simulation run, we first fix the number of buyers and sellers. The seller-side dataset is then distributed uniformly at random across sellers, creating heterogeneous but comparable datasets for different news agencies.

Each buyer is interested in predicting stock price movements on a specific subset of days, corresponding to the days on which the buyer may trade. This induces a buyer-specific prediction task and defines a corresponding test dataset consisting of news articles and stock price indicators for those

days, drawn from the buyer-side dataset originally partitioned for evaluation. To assess the relevance of a seller's dataset for a given buyer, buyer $i$ trains a predictive model $f_{i,j} : \mathbf{x} \mapsto \{0, 1\}$ using only the news headlines provided by seller $j$. The model is implemented as a logistic regression classifier based on term-frequency features extracted from the headlines. The trained model is then evaluated on buyer $i$'s test data, and the variance of its prediction errors is measured. This empirical variance serves as a proxy for the informativeness of seller $j$'s data for buyer $i$'s prediction task, and is used to infer $\tau_{i,j}$, the relevance of seller $j$'s dataset to buyer $i$.

**Dynamics: Inertial Approximate Best Response.** Given the proof of approximate Nash equilibrium in Section 4, we study a natural *inertial response*: each seller $j$ in round $t$, finds the PLC response that guarantees her revenue at least half of her randomized best linear response as defined in Section 4. Recall that this is obtained by solving the following program

$$\max_{\mathbf{x}} \sum_{k=1}^{m_j} x_k \cdot r(\mathbf{e}^{(k)}, \ell_{-j}) \quad \text{subject to} \quad \mathbf{x} \in \Delta_{m_j}.$$

and defining a PLC function $\ell_j^*$, by setting $\ell_{j,k}^* = x_k^*$, where $\mathbf{x}^*$ is a optimal solution to the above program. Then, the seller updates her response in round $t$ as $\ell_j^t$ as $\ell_j^{t-1} \cdot (1 - \alpha) + \ell_j^* \cdot \alpha$, where $\alpha$ is the learning rate controlling the inertia. In our implementation, we set $\alpha = 0.1$, all sellers start with a strategy profile $\ell_{j,k}^0 = 1/m_j$ for every $j, k$, and terminate the dynamics when $\|\ell^{t+1} - \ell^t\|_\infty$ is no more than $\epsilon = 10^{-4}$.

**Convergence Results and Insights.** We evaluated the inertial approximate best-response dynamics on the news data marketplace, varying the number of buyers $n \in \{5, 10, 15, 20\}$ and the number of sellers $m \in \{5, 10, \ldots, 25\}$. Each buyer has a unit budget, and each seller employs a pricing function with five shards and slopes $\{k/10 \mid k \in [10]\}$. As shown in Figure 1a, across all instances the dynamics converge efficiently, terminating within at most 137 rounds.

Although convergence is consistently fast, its speed is not strictly monotonic in the number of sellers. In particular, while the instance with $m = 5$ is smaller than the one with $m = 10$, it requires more iterations to converge. This phenomenon can be attributed to *stronger strategic interdependence in smaller markets*: when the number of sellers is small, a single seller's update induces a larger change in the revenue obtained by other sellers, causing their subsequent responses to vary more significantly and leading to slower stabilization of the dynamics.

As the number of sellers increases, the marginal impact

of any individual seller's update diminishes, reflecting a *large-market* effect. Once this effect becomes prominent, the dynamics stabilize faster, and the convergence rate improves sharply. Beyond this point, we observe a steady and approximately monotone increase in the speed of convergence as the number of sellers continues to grow.

Overall, the empirical results exhibit a non-monotone convergence pattern: convergence is initially slower in very small markets due to strong strategic coupling among sellers, followed by a marked improvement as large-market effects emerge, and finally a smooth and steady growth as the market size increases.

**Approximation Guarantees and Insights.** To quantify the approximation quality of a strategy profile $\boldsymbol{\ell}$, we define its *incentive ratio* as

$$\text{incentive ratio}(\boldsymbol{\ell}) = \max_{j \in [m]} \frac{\max_k r_j(\mathbf{e}^{(k)}, \ell_{-j})}{r_j(\ell_j, \ell_{-j})},$$

which measures the maximum multiplicative gain that any seller can obtain by unilaterally deviating to any linear pricing strategy.

Across all experimental settings, spanning a wide range of buyer and seller counts, we consistently observe that the incentive ratio equals 1. That is, despite our theoretical guarantees establishing only the existence of 2-Nash equilibria, the proposed dynamics reliably converge to *exact* Nash equilibria with respect to linear pricing strategies. Moreover, this phenomenon persists across diverse market sizes, indicating that it is not a fragile artifact of small instances or particular parameter choices.

These findings highlight a gap between worst-case theoretical guarantees and typical empirical behavior, and naturally motivate further theoretical investigation. In particular, they raise the possibility that pricing games arising in real-world data marketplaces exhibit additional structural properties that are not captured by the worst-case analysis, and which may be leveraged to strengthen equilibrium guarantees.

**Sparsity of the PLC Strategies.** A PLC pricing strategy is called *sparse* if it consists of only a few shards with positive length. Sparse strategies are easier to interpret and implement. We measure sparsity as the number of shards with strictly positive length and report the average sparsity of equilibrium pricing functions across market sizes in Figure 1b.

Across all instances, the average sparsity is at most two and always well below the number of buyers or sellers, indicating that equilibrium PLC strategies are nearly linear. Moreover, sparsity increases slightly with fewer sellers, reflecting stronger strategic pricing incentives, and decreases as the number of sellers grows. In particular, when the number of

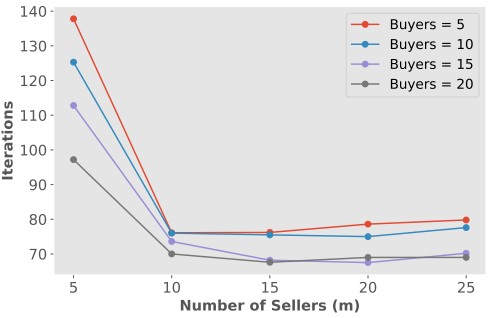

*(a)* Convergence of Inertial Approximate Best Response

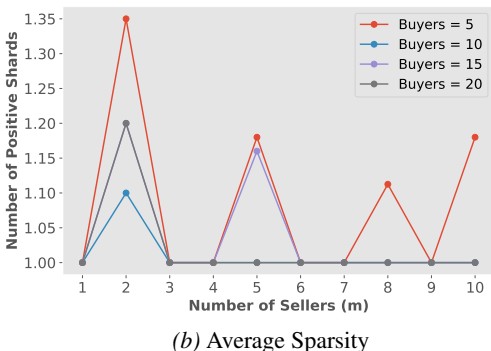

*(b)* Average Sparsity

sellers exceeds two, sellers rarely employ more than a single shard. This decline in sparsity can be attributed similarly to the *large-market effect*: as competition intensifies, the influence of any single seller on market outcomes diminishes, reducing the incentive to employ complex, multi-shard pricing strategies.

## 6. Sparsity of PLC Strategies

Motivated by our empirical observations on sparsity in Section 5, we further study the sparsity of PLC strategies. Formally, seller $j$'s pricing strategy $\ell_j$ is said to be sparse if $|\operatorname{supp}(\ell_j)|$ is small. Sparse strategies are nice because they are easy to represent and communicate.

From the proof of Theorem 4.1, we can infer some structural properties of the 2-approximate Nash equilibrium $\ell^*$ for linear deviations. For any seller $j$, let $B_j$ be the indices of prices that are linear best-responses, i.e.,

$$B_j := \operatorname*{argmax}_{k=1}^{m_j} r_j(\mathbf{e}^{(k)}, \ell_{-j}^*).$$

Then by Lemma 4.2, we get $\operatorname{supp}(\ell_j) \subseteq B_j$. If $B_j$ is small for all $j$, then each seller's pricing strategy is *sparse*, i.e., involving only a few shards. We leave proving an upper bound on $|B_j|$ as an interesting open problem.

We also study the sparsity of PLC best-responses. Specifically, we show that for any strategy profile $\ell$ and any seller $j$, there always exists a PLC best-response $\widehat{\ell}_j$ such that $|\operatorname{supp}(\widehat{\ell}_j)| \leq n$, where $n$ is the number of buyers.

**Lemma 6.1.** *Consider a data market instance $([n], [m], (u_i)_{i=1}^n, (b_i)_{i=1}^n)$. Each seller $j \in [m]$ is given a set $P_j := \{p_{j,1}, \ldots, p_{j,m_j}\}$ of prices, where $p_{j,1} < \ldots < p_{j,m_j}$, and must decide the size $\ell_{j,k}$ of the shard having price $p_{j,k}$.*

*For any seller $j$, given the pricing strategies $\ell_{-j}$ of the other sellers, there is a best-response strategy $\ell_j$ where $|\operatorname{supp}(\ell_j)| \leq n$.*

*Proof.* Fix a seller $j$ and the strategy profile $\ell_{-j}$ of the remaining sellers. We will show how to transform any strategy $\ell_j$ into another strategy $\widehat{\ell}_j$ such that $|\operatorname{supp}(\widehat{\ell}_j)| \leq n$ and seller $j$'s revenue does not decrease, i.e., $r_j(\ell) \leq r_j(\widehat{\ell}_j, \ell_{-j})$.

We will do the transformation in multiple steps. In each step, we apply one of two types of transformations.

**Type 1:** For any $k_1, k_2 \in [m_j]$, the $(k_1, k_2)$-type-1 transformation is applicable when

1. $k_1$ and $k_2$ are adjacent in $\operatorname{supp}(\ell_j)$, i.e., $\ell_{j,k_1} > 0$, $\ell_{j,k_2} > 0$, and $\ell_{j,k} = 0$ for all $k \in [k_1 + 1, k_2 - 1]$.
2. Some buyer $i$ purchases at least part of shard $k_2$.
3. Every buyer $i$ either doesn't buy any part of $k_1$, or finds shard $k_2$ desirable and completely buys every shard (of positive size, and of any seller) in her bang-per-buck order that appears before $k_2$.

In this transformation, we transfer the size of shard $k_1$ to shard $k_2$, i.e., if $\widehat{\ell}_j$ is the new pricing strategy, then $\widehat{\ell}_{j,k_1} = 0$, $\widehat{\ell}_{j,k_2} = \ell_{j,k_1} + \ell_{j,k_2}$, and $\widehat{\ell}_{j,k} = \ell_{j,k}$ for all $k \in [m_j] \setminus \{k_1, k_2\}$. One can see that a $(k_1, k_2)$-type-1 transformation doesn't decrease revenue from any buyer by laboriously listing out the possibilities of where, along her bang-per-buck order, a buyer may stop buying.

**Type 2:** A type-2 transformation is applicable if $|\operatorname{supp}(\ell_j)| > 1$ and the last shard (of positive size) has no revenue. In this transformation, we transfer the size of the last shard to the second-last shard. Formally, let $k_2 := \max(\operatorname{supp}(\ell_j))$ and $k_1 := \max(\operatorname{supp}(\ell_j) \setminus \{k_2\})$. Then we set $\widehat{\ell}_{j,k_1} = \ell_{j,k_1} + \ell_{j,k_2}$, $\widehat{\ell}_{j,k_2} = 0$, and $\widehat{\ell}_{j,k} = \ell_{j,k}$ for all $k \in [m_j] \setminus \{k_1, k_2\}$. One can see that a type-2 transformation doesn't decrease revenue from any buyer.

We apply as many type-1 transformations as possible, and then apply as many type-2 transformations as possible. One can show that no transformations are possible after this. The inapplicability of type-2 transformations implies that in

$\widehat{\ell}_j$, some buyer purchases at least a bit of every shard. The inapplicability of type-1 transformations implies that for any $k_1$ and $k_2$ adjacent in $\mathrm{supp}(\widehat{\ell}_j)$, some buyer purchases (at least a part of) $k_1$ but doesn't purchase everything just before $k_2$. Thus, there must be at least $|\mathrm{supp}(\widehat{\ell}_j)|$ buyers. $\qquad\square$

## Impact Statement

This paper presents work whose goal is to advance the field of Machine Learning. There are many potential societal consequences of our work, none which we feel must be specifically highlighted here.

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
