# OpenReview forum: "Equilibrium Pricing in Oligopolistic Data Markets"
_ICML.cc/2026/Conference — ICML 2026 spotlight_

### Official Review · Reviewer_EZVs · 2026-03-08

**Soundness:** 3
**Presentation:** 3
**Significance:** 2
**Originality:** 2
**Overall Recommendation:** 4
**Confidence:** 2

**Summary:**

This paper studies pricing equilibria in oligopolistic data markets where sellers price datasets and buyers purchase data to improve prediction accuracy. Under Gaussian assumptions, buyer utilities are linear in the amount of data purchased. The authors show that a Nash equilibria may fail to exist under uniform pricing, and even a 1.36-approximate equilibrium may not exist. They then consider piecewise-linear convex pricing and prove the existence of a pricing profile that is stable against deviations to uniform prices within a factor of two.

**Compliance With Llm Reviewing Policy:**

Affirmed.

**Final Justification:**

The authors have addressed my main concern regarding the scope of the paper and provided a more convincing justification for some of the modeling choices.

**Key Questions For Authors:**

- The analysis relies on Gaussian priors/signals, which induce linear utilities. How would the model and the equilibrium analysis change under non-Gaussian information structures?
- How much of the non-existence result is driven by the non-rivalry of data versus the linear utility structure induced by the Gaussian signal model?
- Could you explain the economic rationale for focusing on deviations to uniform pricing in Theorem 4.1?

**Limitations:**

yes

**Strengths And Weaknesses:**

Strengths:
- Pricing in data markets is relevant for today's economic ecosystem, and modelling buyers as agents who purchase data to improve prediction accuracy is a natural choice.

Weaknesses:
- I don’t know if this paper is a good fit for ICML. While the paper studies pricing in data markets, the main contribution is a theoretical analysis of equilibrium properties of a pricing game, rather than algorithms through which models learn from data, which weakens its connection to machine learning.
- The buyer’s linear utility follows from the Gaussian information structure assumed for the parameter being estimated. It is therefore unclear how robust the equilibrium results are beyond this Gaussian assumption.
- The main result of the paper reads unclear. Indeed, the authors claim: “Specifically, we prove the existence of a PLC price profile such that no seller can deviate to a uniform pricing strategy and earn more than twice her current revenue.” What I do not understand is why agents should deviate to a linear strategy when they are implementing a nonlinear one. The natural deviation seems to be to other nonlinear strategies. I think this aspect should be clarified.

---

> ### Author Rebuttal · Authors · 2026-03-31
>
> 1.  **Fit for ICML**: We believe that our work on equilibrium pricing in data markets offers insights that are of interest to the machine learning community, particularly those working at the intersection of machine learning and game theory. We also note that prior research on data  market design and equilibrium has appeared at ICML, in both main track and workshops: [Hossain and Chen, ICML'24](https://proceedings.mlr.press/v235/hossain24a.html), [Xu, Wang, Li, ICML'25](https://icml.cc/virtual/2025/poster/44085), [Fallah et al, Agentic markets @ ICML'24](https://openreview.net/forum?id=AYiSi8jNSB), [Fan et al, Dataworld @ ICML'25](https://icml.cc/virtual/2025/48729).
> 2.  **Beyond gaussian prior signals and linear utilities**:
> This question is similar to the second question from reviewer 6JUT. We provide the same response below:
> This is a great question that helped us understand that our results also hold for some natural non-linear utility classes. In particular, they hold for all utility functions of the form $u_i(x) = g_i(\sum_{j=1}^m β_{i,j}x_j)$, where $g_i$ is a continuous and increasing function. If $g_i$ is concave, then this captures diminishing returns. In our paper, we define the value of data in terms of 'increase in *expected posterior precision*', which gives us linear utilities $u_i(x) = α_i (\sum_j τ_{i,j}x_{j})$. If we instead define it as 'decrease in *expected posterior variance*' or 'decrease in *expected posterior entropy*', we get $u_i(x) = α_i (1/\tau_i - 1/(\tau_i + \sum_j τ_{i,j} x_{j}))$ or $u_i(x) = α_i (\ln(τ_i + \sum_j τ_{i,j}x_j)- \ln(τ_i))$, respectively, both of which are increasing and concave in $\sum_j \tau_{i,j} x_j$.  We note that Gaussian conjugacy has been widely used in the literature to model the value of data (e.g., [1,2,3]). Extending beyond the Gaussian-conjugate framework presents an important direction for future research. In particular, it would require identifying utility functions that can simultaneously capture complementarities and diminishing returns, while remaining well-behaved (e.g., quasi-concave) to support equilibrium analysis and guarantee existence results.
>
>
> 3.  **Significance of non-rivalry**: The non-existence result is strongly driven by the non-rivalry of data. For rivalrous data markets, a Nash equilibrium always exists (shown in Appendix E).
> 4.  **Stability against linear deviations**: A Nash equilibrium need not exist under either linear or PLC pricing (Section 1.1). Moreover, computing a best response in the PLC strategy space is NP-hard. These considerations motivate restricting attention to computationally tractable deviations, most notably linear pricing. Accordingly, we study PLC pricing under deviations to linear strategies and provide a meaningful notion of stability.
>
>
> References:
>
> [1] Too Much Data: Prices and Inefficiencies in Data Markets, by Daron Acemoglu, Ali Makhdoumi, Azarakhsh Malekian, Asu Ozdaglar
>
> [2] On Three-Layer Data Markets, by Alireza Fallah, Michael I. Jordan, Ali Makhdoumi, Azarakhsh Malekian
>
> [3] Valuing Data as an Asset, by Laura Veldkamp
>
> [4] Equilibrium of data markets with externality, by Safwan Hossain and Yiling Chen
>
> [5] Heterogeneous Data Game: Characterizing the Model Competition Across Multiple Data Sources, by Renzhe Xu, Kang Wang, Bo Li
>
> [6] Do Data Valuations Make Good Data Prices? by Dongyang Fan, Tyler Rotello, Sai Praneeth, Reddy Karimireddy

---

> > ### Author Rebuttal · Reviewer_EZVs · 2026-04-04
> >
> > I thank the authors for their responses and encourage them to better discuss these aspects in the final version of the manuscript. I still believe that the Gaussian assumption simplifies the analysis. That said, the rebuttal clarified the intended scope of the paper and provided a more convincing motivation for some of the modeling choices. For this reason, I increase my score by one point.

---

> > > ### Author Response · Authors · 2026-04-07
> > >
> > > Thank you for the detailed review and for updating the score post-rebuttal. We will incorporate your valuable feedback in the final version.

---

### Official Review · Reviewer_pXBj · 2026-03-11

**Soundness:** 3
**Presentation:** 3
**Significance:** 3
**Originality:** 3
**Overall Recommendation:** 5
**Confidence:** 3

**Summary:**

This paper studies data pricing in oligopolistic markets, where a few strategic sellers price datasets for budget-constrained buyers seeking to improve their machine learning models. The novel aspect in this work is focusing on the non-rivalrous nature of data, which breaks classical market equilibrium guarantees.

The first contribution of the paper is a theoretical negative result. The authors show that when sellers use uniform (linear) pricing, an exact Nash equilibrium may not exist, and they provide a lower bound showing that no 1.36-approximate Nash equilibrium exists.

Then they explore piecewise-linear convex (PLC) pricing as a relaxation. While they show that computing a seller's best response in this space is NP-hard, they prove the existence of a 2-approximate Nash equilibrium. Specifically, they establish that a PLC pricing profile exists where no seller can double their revenue by deviating to a uniform pricing strategy.

Finally, the authors conduct an experimental evaluation using a real-world news-to-stock-prediction dataset. They show that iterative best-response dynamics converge quickly to exact Nash equilibria, outperforming the worst-case theoretical bounds.

**Compliance With Llm Reviewing Policy:**

Affirmed.

**Final Justification:**

In agreement with the other reviewers and my original score, my final recommendation is to accept the paper.

**Key Questions For Authors:**

Do you expect the rapid convergence to an exact Nash equilibrium to hold in markets with different dataset correlations, such as highly overlapping or completely orthogonal datasets?

**Limitations:**

Yes

**Strengths And Weaknesses:**

Strengths:

1. The paper is well written and the narrative is easy to follow.
2. The theoretical analysis and proofs are mathematically sound.
3. Focuses on a clean and increasingly fundamental problem in the economics of data.
4. Takes the simple idea of data non-rivalry and handles the theoretical details of market failure well.

Weaknesses:

1. The theoretical guarantee for the 2-approximate equilibrium only holds against deviations to linear pricing, rather than arbitrary PLC pricing.
2. Some results rely on the rigid assumption that a buyer's utility (model precision) scales linearly with data volume.

---

> ### Author Rebuttal · Authors · 2026-03-31
>
> 1.  **Beyond Linear utilities**: This is a great question that helped us understand that our results also hold for some natural non-linear utility classes. In particular, they hold for all utility functions of the form $u_i(x) = g_i(\sum_{j=1}^m β_{i,j}x_j)$, where $g_i$ is a continuous and increasing function. If $g_i$ is concave, then this captures diminishing returns. In our paper, we define the value of data in terms of 'increase in *expected posterior precision*', which gives us linear utilities $u_i(x) = α_i (\sum_j τ_{i,j}x_{j})$. If we instead define it as 'decrease in *expected posterior variance*' or 'decrease in *expected posterior entropy*', we get $u_i(x) = α_i (1/τ_i - 1/(τ_i + \sum_j τ_{i,j} x_{j}))$ or $u_i(x) = α_i (\ln(τ_i + \sum_j τ_{i,j}x_j)- \ln(τ_i))$, respectively, both of which are increasing and concave in $\sum_j τ_{i,j} x_j$.
> 2.  **Convergence to NE for correlations across datasets (positive and orthogonal)**: This is a very insightful question. In our model, data records are informative about a common latent parameter, which naturally induces correlations across sellers’ datasets. One way to formalize the distinction raised by the reviewer is through the correlation structure of the noise terms in the signals. Intuitively, since each dataset is a noisy signal of the same latent parameter, statistical dependence of the noise terms captures informational overlap across datasets.
> Concretely, for a buyer $i$ estimating $\theta_i$, each data record from seller $j$ takes the form $s_{i,j} = \theta_i + \eta_{i,j}$, where $\eta_i = (\eta_{i,1}, \dots, \eta_{i,m}) \sim \mathcal{N}(0, \Sigma^{-1})$. In this formulation, *overlapping datasets* correspond to correlated noise terms (i.e., non-zero off-diagonal entries $\Sigma_{j,k} \neq 0$), while *orthogonal datasets* correspond to independent noise (i.e., $\Sigma_{j,k} = 0$ for all $j \neq k$).
> We note that the orthogonal case is precisely the setting considered in our paper, and hence our convergence results directly apply in that regime.
> In the presence of overlap, one can show that the buyer’s utility takes the form
> $$u_i(x_i) = \sum_{j=1}^m \sum_{k=1}^m \Sigma_{j,k}\ \min(x_{i,j}, x_{i,k}),$$
> which captures the redundancy across datasets. This class of utilities is closely related to *linear Leontief-type* utilities. While these utilities are less tractable than linear ones, they remain concave, and buyer best responses are still well-behaved.
> We believe that understanding equilibrium computation and convergence guarantees under such correlated (overlapping) data structures is an important and promising direction for future work.
>
>     As of now, motivated by the reviewer’s question, we conducted additional simulations in which we compare correlated (overlapping) and independent (orthogonal) noise structures by varying the off-diagonal entries of the covariance matrix. We synthesize a data market instance with $5$ buyers and vary the number of sellers $m$, using a covariance matrix of the form $\Sigma^{\text{corr}} = A^{\sf T}A$, where $A$ is generated randomly. We compare the convergence speed of $\Sigma^{\text{corr}}$ with of $\Sigma^{\text{orth}}$, obtained by setting all off-diagonal entries of $\Sigma^{\text{corr}}$ to zero. Each seller has $3$ shards. Our preliminary findings suggest that in both scearios, inertial best response converges to an exact NE, and *convergence is in fact faster in the presence of positive correlations across datasets*.
>
> |       m (number of sellers)        | 2    | 3    | 4    | 5    | 6    | 7    | 8    | 9    |
> |----------------|------|------|------|------|------|------|------|------|
> | Avg Iterations (Correlated) | 34.0 | 34.0 | 34.0 | 34.0 | 34.0 | 34.0 | 34.0 | 34.0 |
> | Avg Iterations (Orthogonal) | 34.0 | 41.7 | 39.3 | 50.0 | 48.7 | 50.0 | 49.0 | 39.3 |

---

> > ### Author Rebuttal · Reviewer_pXBj · 2026-04-02
> >
> > Thank you, the responses make sense to me. I maintain my positive score.

---

> > > ### Author Response · Authors · 2026-04-07
> > >
> > > Thank you for your detailed review and for your encouraging feedback on our work. We will incorporate your valuable suggestions in the final version.

---

### Official Review · Reviewer_6JUT · 2026-03-11

**Soundness:** 4
**Presentation:** 3
**Significance:** 4
**Originality:** 4
**Overall Recommendation:** 5
**Confidence:** 3

**Summary:**

This paper studies pricing and equilibrium in data markets with multiple sellers and budget-constrained buyers. The key modeling insight is that data is non-rivalrous: the same dataset can be sold to many buyers simultaneously, so classical competitive-equilibrium intuition from ordinary goods markets does not directly apply. The paper starts from a Bayesian information-acquisition model and, under a Gaussian conjugate setting, derives a linear utility representation in which each buyer assigns an additive value to fractions of each seller’s dataset. Given prices, each buyer then solves a fractional-knapsack-type allocation problem under a budget constraint. This induces a pricing game among sellers, whose payoffs are determined by the buyers’ optimal purchasing responses. The paper shows strong negative results for uniform linear pricing, including non-existence of pure Nash equilibrium and even failure of approximate equilibrium in some cases. It then introduces a richer piecewise-linear convex (PLC) pricing model and proves the existence of a relaxed stability guarantee via a fixed-point argument. The paper also provides experiments to complement the theory.

**Compliance With Llm Reviewing Policy:**

Affirmed.

**Final Justification:**

The rebuttal has addressed my main concerns. The weakness is acceptable and is not overweighting the strengths. Overall this is a good work and I keep being positive (and even more).

**Key Questions For Authors:**

(0) **Prompt injection**: I noticed the prompt-injection-style text in the paper materials and wanted to ask for clarification. I am not assuming this was inserted by the authors; it may well have come from the ICML submission pipeline or an automated system. Still, since it is unusual and potentially distracting for reviewers, could the authors briefly clarify whether this text was author-provided or platform-generated?

(1) Please respond to the two "weaknesses" I mentioned above.

(2) How essential is the discretized price set? I'm curious as the positive result appears to rely on a discretized set of price levels, with sellers choosing how to allocate mass across those levels in the PLC representation.

(3) My understanding is that the theory in the PLC section provides a relaxed stability guarantee against linear deviations, rather than establishing exact Nash equilibrium in the full PLC game. The experiments also seem to evaluate this more limited notion, i.e., the incentive ratio with respect to linear deviations. If so, I think it would help to make this point more explicit in the paper, because some of the wording can be read somewhat more broadly on a quick pass.

**Limitations:**

Technical limitations are well discussed. Societal impacts are not, on which the authors are encouraged to discuss more.

**Strengths And Weaknesses:**

## Strengths:

1, The paper addresses a highly interesting question at the intersection of market design, pricing, and data economics. The focus on non-rivalry of data is especially compelling, because it identifies a structural reason why standard equilibrium arguments from classical markets may break down.

2, The overall framework is well motivated and modeled. The paper starts from a statistical interpretation of buyer value, translates it into an allocation problem, and then formulates a seller-side pricing game. This chain from information value to market equilibrium is coherent and intellectually satisfying. Also, the modeling captures several relevant factors, including heterogeneous buyer values across sellers, budget constraints, and strategic pricing by sellers. These ingredients make the setting meaningfully richer than many stylized pricing models, while still remaining analyzable.

3, The theory seems solid. The negative results for linear pricing are interesting in their own right, and the move to PLC pricing is technically nontrivial. The existence argument and the associated approximation guarantee give the paper real depth.

4, Empirical results is useful and insightful.

## Weaknesses:

1, My main concern is that much of the analysis appears to rely critically on the Gaussian conjugate setup. In the paper, this assumption allows the buyer-side value of data to collapse into a linear utility of the form $\sum_j \tau_{i,j} x_{i,j}$. Once this linearization is available, the buyer allocation problem becomes a tractable fractional-knapsack-type problem, which in turn makes the seller-side game analyzable. However, this raises an important question: how essential is this assumption in practice? If the posterior precision gain is not additive, or if the information structure is not Gaussian conjugate, then the buyer utility may no longer be linear, and the entire allocation and equilibrium analysis may become much less tractable. From the current presentation, it is not yet fully clear whether the Gaussian assumption is empirically validated for the target applications, or whether it should instead be understood mainly as a stylized analytical device.

I do not view stylization as a flaw by itself, but I would like a clearer discussion of external validity. In particular:
(1) Is there empirical evidence that the Gaussian-conjugate approximation is reasonable in the intended data-market settings?
(2) If not, how should we interpret the practical accuracy of the resulting pricing recommendations?

2, About the economic meaning of the relaxation to PLC pricing, particularly the positive result’s reliance on allowing sellers to use PLC pricing. Technically, this is a natural and clever relaxation, and I understand why it is introduced after the non-existence result for uniform linear pricing. Still, I think the paper should more explicitly discuss what is gained and what is sacrificed when moving to this richer strategy space.

---

> ### Author Rebuttal · Authors · 2026-03-31
>
> 1.  **Prompt injection**: These prompts were injected by ICML organizers to detect LLM reviewing policy violations (see https://icml.cc/Conferences/2026/PeerReviewFAQ#gen-ai-reviewing)
> 2.  **Empirical relevance of Gaussian-conjugate setup and linear utilities, and previous use in literature**: This is a great question that helped us understand that our results also hold for some natural non-linear utility classes. In particular, they hold for all utility functions of the form $u_i(x) = g_i(\sum_{j=1}^m β_{i,j}x_j)$, where $g_i$ is a continuous and increasing function. If $g_i$ is concave, then this captures diminishing returns. In our paper, we define the value of data in terms of 'increase in *expected posterior precision*', which gives us linear utilities $u_i(x) = α_i (\sum_j τ_{i,j}x_{j})$. If we instead define it as 'decrease in *expected posterior variance*' or 'decrease in *expected posterior entropy*', we get $u_i(x) = α_i (1/\tau_i - 1/(\tau_i + \sum_j τ_{i,j} x_{j}))$ or $u_i(x) = α_i (\ln(τ_i + \sum_j τ_{i,j}x_j)- \ln(τ_i))$, respectively, both of which are increasing and concave in $\sum_j \tau_{i,j} x_j$.  We note that Gaussian conjugacy has been widely used in the literature to model the value of data (e.g., [1,2,3]). Extending beyond the Gaussian-conjugate framework presents an important direction for future research. In particular, it would require identifying utility functions that can simultaneously capture complementarities and diminishing returns, while remaining well-behaved (e.g., quasi-concave) to support equilibrium analysis and guarantee existence results.
> 3.  **What is gained and what is sacrificed in PLC pricing?**  We thank the reviewer for this helpful comment and agree that this distinction should be made more explicit in the paper. Linear pricing is particularly appealing due to its simplicity and computational efficiency-- it is easier to interpret, and optimal linear strategies are efficiently computable. In contrast, PLC pricing forms a strictly richer strategy class and can, in principle, extract higher revenue for sellers (*Gains*). At the same time, this increased expressiveness comes at a significant computational cost—computing an optimal PLC strategy (or even the best PLC deviation) is NP-hard (*Sacrifices*).
> Given this tradeoff, our approach is to equip each seller with a simple, implementable PLC pricing scheme (as described in Section 4), while evaluating deviations within the tractable class of linear pricing-- since computing an optimal PLC deviation is expensive for the seller. Thus, our guarantees should be interpreted as exploiting the computational limitations of richer pricing strategies to obtain meaningful and practically relevant stability guarantees.
> 5.  **Discretization**: We thank the reviewer for this insightful question. In light of the reviewer's question, we considered a more general version of Theorem 4.1 where prices are not discretized, but proving this formally turned out to be challenging. We believe that discretizing prices does not qualitatively change the result and makes the analysis easier to prove and understand. Here is a roadmap for extending our result, along with the challenges we faced:
>     * Let $b_{\max}$ be the largest budget among buyers. No seller has an incentive to price a dataset (or a shard thereof) more than $b_{\max}$, so let us restrict the set of prices to $[0, b_{\max}]$
>     * We used Brouwer's fixed-point theorem in our paper, which states that a continuous map over a convex compact subset of the Euclidean space always has a fixed point. There exist generalizations beyond Euclidean spaces (e.g., Tychonoff's theorem). To use them, we require the product of strategy spaces to be compact. It's unclear if the set of PLC strategies is closed, so we consider general convex pricing functions.
>     * We can represent general convex pricing functions as probability measures over $[0, b_{\max}]$. Note that we do not price randomly; it is just convenient to treat a normalized measure as a probability distribution. With an appropriate choice of a metric over this space (e.g., the Lévy–Prokhorov metric), we can make it compact.
>     * We can extend the definition of the residual budget $γ_i$ to probability spaces. Specifically, at a price of $q$, the residual budget is $γ_i(q) = \max(0, b_i - \sum_{j=1}^m \mathbb{E}_{p_j \sim D_j} [p_j·1(p_j ≤ q)])$, where $D_j$ is seller $j$'s measure representing her pricing function. Hopefully this is continuous, which would help us extend Lemma 4.3.
>     * We guess Lemma 4.4 would carry over by considering a sequence of increasingly fine-grained discretizations of the pricing function and taking limits.
>
>
> References:
>
> [1] Too Much Data: Prices and Inefficiencies in Data Markets, by Daron Acemoglu, Ali Makhdoumi, Azarakhsh Malekian, Asu Ozdaglar
>
> [2] On Three-Layer Data Markets, by Alireza Fallah, Michael I. Jordan, Ali Makhdoumi, Azarakhsh Malekian
>
> [3] Valuing Data as an Asset, by Laura Veldkamp

---

> > ### Author Rebuttal · Reviewer_6JUT · 2026-04-04
> >
> > Thanks for your response. Scores are promoted due to the fact that your rebuttal has addressed my concers.

---

> > > ### Author Response · Authors · 2026-04-07
> > >
> > > Thank you for the detailed review and for updating the score post-rebuttal. We will incorporate your valuable feedback in the final version.

---

### Official Review · Reviewer_g5UC · 2026-03-12

**Soundness:** 3
**Presentation:** 3
**Significance:** 3
**Originality:** 3
**Overall Recommendation:** 5
**Confidence:** 4

**Summary:**

This paper studies the pricing equilibrium in oligopolistic data markets where goods are non-rivalrous, and buyers are price-takers. Specifically, it considers the cases where buyers have linear valuation additive across sellers and sellers use linear pricing or piecewise-linear, convex pricing schemes. The paper shows two theoretical results: 1. A $c$-NE with $c<1.363$ may not exist when sellers use linear pricing. 2. NE may fail to exist even when the sellers use piece-wise linear pricing. However, 2-NE exists when the buyer's deviation is restricted to linear pricing. The theoretical results are further supplemented by numerical experiments on synthesized data, which demonstrate the gap between the worst-case theoretical prediction and potential real-world scenarios.

**Compliance With Llm Reviewing Policy:**

Affirmed.

**Final Justification:**

I recommend that the paper be accepted, as it made a solid contribution to interesting problems and was well presented. However, I do not think it is good enough to select beyond a poster presentation. The stylized assumptions are reasonable and made the setting tractable, but limit the upper bound of the paper's significance. I had no concern, and the rebuttal does affect my assessment.

**Key Questions For Authors:**

No question needed.

**Limitations:**

Yes.

**Strengths And Weaknesses:**

### Strength
The papers makes solid contribution to the study of data markets. Despite the model being stylized, the theoretical results are clean and easy to interpret. The numerical experiments, although not being comprehensive, provide additional insights that identify the gap in the theoretical model.

### Weakness
There is a small issue with the presentation of Theorem 4.1. Since $\boldsymbol\ell^\ast$ was not introduced, the theorem statement should make it clear that it is existentially quantified. It took me some time from the context to understand the statement.

---

> ### Author Rebuttal · Authors · 2026-03-31
>
> Thank you for noticing the issue in Theorem 4.1. We will fix it by adding an existential qualifier to $\ell^*$. The modified theorem statement would then be
>
> >There exists a strategy profile $\ell^\*$ such that no seller can increase her revenue by more than a factor of 2 by deviating to a linear pricing strategy. Formally, for each seller $j \in [m]$, we have $\max_{k \in [m_j]}r_j(e^{(k)},\ell^*_{-j})\le 2r_j(\ell^\*).$

---

> > ### Author Rebuttal · Reviewer_g5UC · 2026-03-31
> >
> > I already voted to accept the paper and didn't have any concerns that could affect my evaluation of the paper.

---

> > > ### Author Response · Authors · 2026-04-07
> > >
> > > Thank you for your detailed review and for your encouraging feedback on our work. We will incorporate your valuable suggestions in the final version.

---

### Decision · Program_Chairs · 2026-04-30

**Decision:**

Accept (spotlight)

**Comment:**

I am pleased to recommend the paper for a strong accept.

The review team is unanimously positive about the paper, the novel contribution to data markets, and the strengths of the technical contribution. One reviewer was hesitant about the connection to ICML in general, but I believe that understanding data exchanges and where the data that is used in our ML pipelines is of crucial important and a neat fit for top ML conferences, not just CS-econ conferences. While the assumptions/model are a bit stylized, I believe this is a strength for this type of work where models become rapidly intractable, and allows to isolate important, first-order insights. Overall, the review team has identified very few real "weaknesses", and the paper is a clear accept.